# Antibacterial, Trichomonacidal, and Cytotoxic Activities of *Pleopeltis crassinervata* Extracts

**DOI:** 10.3390/pharmaceutics16050624

**Published:** 2024-05-07

**Authors:** Jhony Anacleto-Santos, Elisa Vega-Ávila, Leticia Pacheco, Manuel Lacueva-Arnedo, Alicia Gómez-Barrio, Alexandra Ibáñez-Escribano, Teresa de Jesús López-Pérez, Brenda Casarrubias-Tabarez, Fernando Calzada, Perla Yolanda López-Camacho, Norma Rivera-Fernández

**Affiliations:** 1Departamento de Microbiología y Parasitología, Facultad de Medicina, Universidad Nacional Autónoma de México (UNAM), Ciudad Universitaria, Mexico City 04510, Mexico; tere.lopez82@comunidad.unam.mx; 2Departamento de Ciencias de la Salud, Universidad Autónoma Metropolitana Iztapalapa, Mexico City 09340, Mexico; vega@xanum.uam.mx; 3Departamento de Biología, Universidad Autónoma Metropolitana Iztapalapa, Mexico City 09340, Mexico; pacheco@xanum.uam.mx; 4Departamento de Microbiología y Parasitología, Facultad de Farmacia, Universidad Complutense de Madrid, 28040 Madrid, Spain; mlacueva@ucm.es (M.L.-A.); agbarrio@ucm.es (A.G.-B.); alexandraibanez@ucm.es (A.I.-E.); 5Departamento de Biología Celular y Tisular, Facultad de Medicina, Universidad Nacional Autónoma de México (UNAM), Ciudad Universitaria, Mexico City 04510, Mexico; bcasarrubias@facmed.unam.mx; 6Unidad de Investigación Médica en Farmacología, Unidad Médica de Alta Especialidad, Hospital de Especialidades Centro Médico Nacional Siglo XXI, Instituto Mexicano del Seguro Social, Col. Doctores, Cuauhtémoc, Mexico City 06725, Mexico; fernando.calzada@imss.gob.mx; 7Departamento de Ciencias Naturales, Universidad Autónoma Metropolitana Cuajimalpa, Mexico City 05370, Mexico; plopezc@correo.cua.uam.mx

**Keywords:** traditional medicine, antibacterial activity, trichomonacidal effect, *Pleopeltis crassinervata*

## Abstract

*Pleopeltis crassinervata* is a fern documented in ethnobotanical records for its use in Mexican traditional medicine to treat gastric disorders and mouth ulcers. Consequently, conducting biological and pharmacological assays is crucial to validate the therapeutic efficacy of this plant within the context of traditional medicine. In the present study, we investigated the biological activity of extracts and fractions obtained from *P. crassinervata* organs against bacteria (*Salmonella typhimurium*, *Salmonella typhi*, *Staphylococcus aureus*, *Proteus mirabilis*, *Shigella flexneri*, *Bacillus subtilis*, *Escherichia coli)* and *Trichomonas vaginalis* using in vitro models. The precipitate fraction obtained from the frond methanolic extract showed significant antibacterial activity (minimal inhibitory concentration [MIC] 120 µg/mL) against the *Staphylococcus aureus* strain and was effective against both Gram-positive and Gram-negative bacteria. The hexane fraction also obtained from frond methanolic extract, showed a trichomonacidal effect with an IC_50_ of 82.8 μg/mL and a low cytotoxic effect. Hsf6 exhibited the highest activity against *T. vaginalis*, and the GC-MS analysis revealed that the predominant compound was 16-pregnenolone. The remaining identified compounds were primarily terpene-type compounds.

## 1. Introduction

In recent years, the importance of natural products research has surged with a particular emphasis on utilizing ethnobotanical knowledge to guide the selection of medicinal plants for thorough biological assessment [1]. In this context, Mexico, a megadiverse nation, emerges as a prominent player ranking among the top three of the World’s 12 megadiverse countries in terms of natural resources. Mexico holds a 10–12% share of global biodiversity, nurturing a diverse array of 21,989–23,424 vascular plant species. Remarkably, it holds the second position worldwide for its extensive array of medicinal plant species, totaling approximately 4500, of which only approximately 5% have been evaluated in chemical-pharmacological research [2]. It is worth highlighting that approximately 90% of Mexico’s population relies on these medicinal plants to alleviate basic health issues [3]. This intricate interplay between Mexico’s rich biodiversity and cultural reliance on medicinal flora underscores the importance of exploring the country’s botanical resources.

Ferns, classified within the Pteridophyte division, have historically received limited attention in pharmacological research. Nonetheless, it is noteworthy that compounds such as flavonoids, terpenoids, alkaloids, and glycosides have been identified within this diverse group of plants [4,5].

*Pleopeltis crassinervata*, known locally as “lengua de ciervo,” holds medicinal significance in Querétaro, Mexico, where its frond infusions are used to treat gastric disorders and mouth ulcers [6]. While taxonomic, morphological, and ecological studies of *P. crassinervata* have been conducted, further research is urgently needed, particularly in biological and pharmacological assays to validate its traditional medicinal applications.

Over the last four decades, from 1981 to 2019, an intriguing phenomenon has been observed in the pharmaceutical field. Of the 1881 drugs approved during this period, 41.9% originated from natural products. Despite this, a worrying disparity is evident: only 9 of these drugs correspond to antiparasitics, and 93 to antibacterial medications, highlighting a notable lack of research within these specific areas [7]. In parallel, the possibility arises that active molecules may exhibit dual properties, taking advantage of shared characteristics among groups of organisms that are not taxonomically related. This manifests in metabolic similarities, such as protozoa and bacteria which share the folic acid synthesis pathway. This metabolic similarity has prompted extensive research focused on antifolates. These compounds interfere with the folic acid synthesis pathway, an essential route for the survival of these organisms. Notable examples of these compounds include pyrimethamine, sulfonamides, trimethoprim, and dapsone [8].

Bacterial infections are widespread worldwide, and the emergence of resistant strains poses a significant public health issue, even in developed countries. Through the Global Antimicrobial Resistance Surveillance System (GLASS) the World Health Organization (WHO) collects valuable data on antibiotic resistance, including the resistance mechanisms to identify effective alternatives for treating these infections [9,10]. Currently, according to the WHO, antibiotic resistance is a major public health challenge that involves extended hospital stays and increased mortality rates. Approximately 700,000 deaths are attributed to complications arising from antibiotic resistance each year [11,12,13].

In contrast, trichomoniasis, a parasitic infection caused by the protozoan *Trichomonas vaginalis*, is a globally prevalent sexually transmitted disease, affecting up to 23% of the population worldwide predominantly women [14]. The latest WHO report estimates an incidence of more than 156 million cases in 2018 [15]. Although 50–75% of *T. vaginalis* cases are asymptomatic, complications include pregnancy outcomes, pelvic inflammatory disease (PID), and an increased risk of acquiring other STIs and developing cervical or prostate cancer [16,17]. There is only one family of drugs approved for use against *Trichomonas*: the 5-nitroimidazoles [16]. Nevertheless, there is a growing demand for safer and more effective alternatives, given the resistance to currently available pharmacological treatments and the need to minimize side effects [18]. Recent investigations indicate that the hexanic fraction from *P. crassinervata* affects the viability of *Toxoplasma gondii* at low concentrations with an IC_50_ 16.9 µg/mL [19].

Based on the antimicrobial properties coupled with the antiparasitic activity exposed in previous studies, evaluations of extracts, fractions, and subfractions of the plant were carried out through a bioguided approach [19,20]. In vitro models with pathogenic bacteria, including both Gram-positive and Gram-negative strains that are commonly associated with gastric disorders and mouth ulcers, were employed for these investigations. Furthermore, we explored the activity of *P. crassinervata* against the protozoan *T. vaginalis*. Cytotoxicity tests were also performed to ensure the safety of these natural products. This comprehensive research approach aims to identify natural products in the plant that can be utilized to develop safe and effective experimental treatments against bacterial and parasitic infections.

## 2. Materials and Methods

### 2.1. Bacterial Strains

*Salmonella typhimurium* ATCC 14028, *Salmonella typhi* ATCC 6539, *Staphylococcus aureus* ATCC 6538, *Proteus mirabilis* NTC 289, *Shigella flexneri* ATCC 29003, *Bacillus subtilis* ATCC6633, and *Escherichia coli* ATCC 8739, obtained from the strain collection of Escuela Nacional de Ciencias Biológicas del Instituto Politécnico Nacional, Mexico, were inoculated in Mueller-Hinton 3× medium at 37 °C.

### 2.2. Parasites

*Trichomonas vaginalis* trophozoites JH31A#4 (ref. 30326 from de American Type Culture Collection (ATCC), Gaithersburg, MD, USA) were cultured in modified Tryptone Yeast Maltose medium (TYM) at pH 6 with 10% (*v*/*v*) heat-inactivated fetal bovine serum (FBS), and antibiotic solution (100 UI/mL of penicillin and 100 μg/mL of streptomycin). The parasites were incubated at 37 °C and 5% CO_2_ and sub-cultured every 48–72 h.

### 2.3. Cell Culture

To evaluate the unspecific cytotoxic effect, African green monkey kidney epithelial cells (Vero CCL-81) maintained in Roswell Park Memorial Institute medium (RPMI-1640) supplemented with 10% inactive FBS at 37 °C under 5% CO_2_ atmosphere were used.

### 2.4. Plant Material, Extraction, and Fractionation

*P. crassinervata* was collected during the reproductive period (September–October) in Chignautla, Puebla, Mexico (19°50’19.52″ N 97°22’29.60″ W). The plant specimen was identified and authenticated by Leticia Pacheco (taxonomist) and deposited in the Metropolitan herbarium Dr. Ramón Riba and Nava Esparza (UAMIZ), CDMX, voucher number 84415. The plants were separated into fronds, roots, and rhizomes, dried at 25 °C, and ground. One hundred grams of dried and ground vegetal material from each organ of *P. crassinervata* was subjected to extraction through maceration with solvents (1:10 *w*/*v*) for 4 weeks at room temperature. Water, methanol, and hexane were employed as solvents. The extracts were filtered and concentrated in a rotary evaporator to remove the solvent. A second frond extraction was carried out as in the first antibacterial in vitro studies, this extract showed the highest biological activity. The extraction was carried out using a solid–liquid–liquid continuous system (modified Soxhlet) for 7 days with methanol, employing dried and ground fronds of *P. crassinervata* (1:10 *w*/*v*) at 30 °C. Finally, the extracts were filtered, concentrated, lyophilized, and stored until use. The lyophilized methanolic frond extract (FMeOHe) was dissolved in methanol (1:10 *w*/*v*), and an equivalent volume of hexane was added to the solution. Three fractions were obtained (hexane, methanolic, and precipitate), the solvents were vacuum-evaporated, and the fractions were lyophilized. Due to its biological efficacy, hexane fraction (Hf) was subjected to a chromatographic separation in a glass column packed with gel silica, and seven subfractions were obtained. The subfractions were named Hsf1–Hsf7 in accordance with the elution order, as described in our previous study [20].

### 2.5. Antibacterial Assay

The antibacterial activity was assessed by measuring the minimum inhibitory concentration (MIC) using the resazurin method as an indicator of viability [21]. The resazurin dye undergoes reduction to resorufin by oxidoreductase enzymes, which are only active in living cells. This results in a color change from blue to pink.

An antibacterial evaluation was performed from stock solutions of the extract in Dimethyl sulfoxide (DMSO), and a series of double dilutions were made in eight consecutive wells in triplicate on a 96-well microplate, using physiological saline solution as a solvent. The range of concentrations used in the first and second assays was 10–1500 μg/mL and 7.81–1000 μg/mL, respectively. Fifty microliters of each sample (extracts and fractions) diluted in 10 μL of the viable bacteria suspension (4 × 10^6^ CFU/mL), 10 μL of sodium resazurin (0.675% *w*/*v* in physiological saline solution (PSS)), and 30 μL of culture medium Mueller-Hinton 3× were placed in each well. The microplates were incubated at 37 °C for 22 h. DMSO and PSS served as negative controls and a penicillin/streptomycin mixture was used as a positive control [22]. Three independent experiments were performed, and the MIC was ascertained. The MIC corresponds to the lowest concentration of the extract or fraction where the color change occurs [21].

### 2.6. Trichomonacidal Assay

The trichomonacidal activity of the methanolic extract of fronds, their fractions, and subfractions was determined using *T. vaginalis* trophozoites and compared with an untreated growth control group [23]. In vitro trichomonacidal activity tests were carried out on Pyrex^®^ glass tubes containing 1 × 10^5^ trophozoites/mL. After 5 h the exponential growth phase cultures were incubated with evaluated samples in a concentration range of 6.25–200 μg/mL and 3.12–100 μg/mL, for 24 h at 37 °C and 5% CO_2_ according to the solubility of each sample [24,25,26].

The stock solutions were dissolved in DMSO. Metronidazole (MTZ) was used as the reference drug [1, 2, and 4 μg/mL]. Final DMSO concentration in cultures did not exceed 0.1% (*v*/*v*). Thereafter, the cultures were transferred to 96-well microplates, centrifuged at 300× *g* and resuspended in PBS resazurin dye (3 mM stock solution). After 1 h of incubation trichomonacidal effect was determined by fluorometry (λ excitation 535 nm and λ emission 590 nm). Each concentration was evaluated in triplicate, and the values were obtained from the mean of three independent evaluations. In all the assays, a blank control without trophozoites was included. The active fraction IC_50_ was obtained through a linear regression analysis by plotting the extract concentration versus % inhibition.

### 2.7. Cytotoxic Assay

To assess the unspecific cytotoxicity, Vero CCL-81 cell cultures (1 × 10^5^ cells/well) were added in 96-well microplates in Roswell Park Memorial Institute medium (RPMI) + 10% Fetal Bovine Serum (FBS) and then exposed to 6.25–800 µg/mL of *P. crassinervata* Hf subfractions for 24 h at 37 °C and 5% CO_2_. After incubation, 20 µL of resazurin solution 1 µM was added to each well. The microplates were incubated at 37 °C and 5% CO_2_ for 3 h. DMSO in RPMI (0.1% *v*/*v*) was used as the negative control. The viability percentage of Vero cells was determined using fluorometry through a trichomonacidal assay as previously described [23]. The assay was conducted in triplicate on three independent experiments.

### 2.8. Screening for Synergistic Effects of the Fractions against T. vaginalis

To comprehensively assess the effect of the active subfractions combined when present together, an in vitro screening against *T. vaginalis* was performed following the method previously described in the trichomonacidal assay. Specifically, Hsf5 and Hsf6 were subjected to evaluation owing to their activity against the parasite.

For Hsf5, varying concentrations were introduced within the range of 6.25–200 µg/mL, including a fixed IC_50_ concentration of Hsf6 in each well. Similarly, the reverse scenario was executed wherein Hsf6 was introduced at varying concentrations, and a fixed IC_50_ concentration of Hsf5 was introduced alongside it in each well. A fixed concentration of MTZ was also evaluated in combination with the subfractions.

### 2.9. GC/MS Analysis

The identification of organic compounds within Hsf6 was conducted using gas chromatography (Agilent 6890 Plus, Santa Clara, CA, USA) and mass spectrophotometry (Agilent 5973N; GC-MSD, Santa Clara, CA, USA) systems. The GC–MS analysis was outsourced to the Chemical Research Center at the Autonomous University of Morelos (UAEM), Cuernavaca, Morelos, Mexico. Following the evaporation of the solvent under vacuum conditions, 4 mg of Hsf6 was stored and protected from light until needed. The analysis was performed under specific conditions including a flow rate of 1 mL/min in the split-less injection (1 µL) mode, an inlet temperature of 40 °C for 10 min, with interface temperature of 250 °C. Capillary column (Agilent J&W, Wilmington, DE, USA 30 m × 0.25 mm, 0.10 μm) was used and helium was employed as the carrier gas in a 1 h run. The compounds in Hsf6 were identified by comparing the GC–MS data including retention time (min), peak area, and mass spectral patterns, with that of the mass spectral library NIST 1.7a.

### 2.10. Statistical Analysis

The obtained data were analyzed via one-way analysis of variance using the GraphPad Prism^®^ software version 7 (https://www.graphpad.com/, accessed on 18 January 2024). All analyzed data had a normal distribution, and statistical significance was set at *p* < 0.05. Half minimal inhibitory concentration was calculated using a nonlinear regression analysis, and viability percentages were expressed as mean ± standard error of three replicates.

## 3. Results

### 3.1. Antibacterial Assay

The hexane root extract was not evaluated owing to its low solubility in DMSO. The *Staphylococcus aureus* strain was impacted by FMeOHe and rhizome methanolic extract (RMeOHe) exhibiting an MIC value of 373 µg/mL. *Salmonella typhi* and *Shigella flexneri* strains were affected by the methanolic and aqueous extracts of the three organs with MICs of 745 and 1500 µg/mL, respectively. Four of the seven bacterial strains evaluated were affected by *P. crassinervata* extracts, particularly FMeOHe and RMeOHe with the lowest MICs in this evaluation (Table 1). These results agree with the ethnobotanical record of *P. crassinervata* which reported that the frond harbors curative properties [6]. For these reasons we decided to perform the fractionation of FMeOHe according to the methodology established in our previous research, with the aim of elucidating its antibacterial activity [20].

After FMeOHe partition, a second antibacterial assay was performed wherein FMeOHe and its three fractions were evaluated. The most active fraction was the precipitate (with an MIC of 125 μg/mL on *S. aureus*). *P. mirabilis*, *S. typhi*, *S. typhimurium*, and *S. flexneri* were also affected with MIC values between 500 and 1000 μg/mL (Table 2). MIC concentrations < 100 μg/mL are considered highly effective in the field of natural products research with antibacterial activity [27]. To date, there have been no scientific reports on *P. crassinervata* extracts activity on *S. typhi* and *S. flexneri* strains. It is worth noting that none of the fractions exhibited activity against *Bacillus subtilis* and *Escherichia coli* strains.

### 3.2. Effect of P. crassirnervata in T. vaginalis Trophozoites and Cell Viability

The Hf was evaluated at concentrations of 3–100 μg/mL, and FMeOHe and its methanolic and precipitate fractions at 6–200 μg/mL. FMeOHe showed a low trichomonacidal activity with an inhibition percentage of 7.87% at the maximum evaluated concentration. The methanolic and precipitated fractions showed inhibition percentages of 28.26 and 33.2%, respectively, at 200 μg/mL. The Hf depicted the highest activity (52%) at 100 μg/mL with an IC_50_ of 82.3 μg/mL (Table 3). The Hf subfractions (Hsf) obtained through column chromatography were assessed under identical conditions. This process yielded seven distinct subfractions, each labeled sequentially based on their elution order. Notably, as this is an indirect assay for determining cell viability, the initial data exhibited fluctuations due to the stabilization of the resorufin dye. The fluctuations in the data stabilized once the concentrations reached 50 µg/mL, allowing for clear differentiation between the treated groups. Among these subfractions, Hsf5 and Hfs6 emerged as the most potent, as evidenced by the results of a nonlinear regression analysis, which revealed IC_50_ values of 90.9 µg/mL and 105.4 µg/mL, respectively.

Subfractions Hsf1–4 displayed negligible trichomonicidal activity failing to demonstrate any statistically significant impact on the parasites (Figure 1). Notably, Hsf7 exhibited a modest activity against the parasites, with values reaching up to 26% at the highest evaluated concentration of 200 µg/mL.

Based on the screening results against *T. vaginalis*, the cytotoxicity assay was conducted in Vero cells with the two most effective extracts (Hsf5 and Hsf6). These assessments encompassed a range of concentrations from 6.2 to 800 µg/mL. The findings revealed that, at concentrations up to 200 µg/mL, none of the subfractions exhibited any discernible cytotoxic effects. However, a concentration-dependent cytotoxic effect became evident as the concentrations increased beyond this threshold. At the highest concentration tested, Hsf5 and Hsf6 demonstrated cytotoxicity levels of up to 42.5% and 74.7%, respectively.

Subsequently, employing a nonlinear regression analysis, the half cytotoxic concentration (CC_50_) values were calculated for Hsf5 and Hsf6, yielding values of 394.3 µg/mL and 416.6 µg/mL, respectively. Furthermore, a selectivity index of 4.37 and 3.91 was determined for Hsf5 and Hsf6, respectively, shedding light on the relative safety of these subfractions in the context of cytotoxicity when compared to their trichomonacidal activity (Figure 2).

### 3.3. Synergistic Effect of P. crassinervata Trichomonacidal Subfractions

Based on the data obtained from the pharmacological screening of *P. crassinervata* subfractions against *T. vaginalis*, the active subfractions exhibited lower activity than the hexane fraction. Therefore, to demonstrate the synergistic effect of these subfractions, a trichomonacidal activity test was conducted using Hsf5 and Hsf6 within a concentration range of 6.2–200 µg/mL, supplemented with a fixed concentration of MTZ (IC_50_ 5 µg/mL) [25]. In another experiment, Hsf5 was supplemented with a fixed IC_50_ concentration of Hsf6, and vice versa. Hsf1 and Hsf6 when supplemented with a fixed concentration of MTZ, impacted the cultures with an antiparasitic effect near 80% at a concentration of 100 µg/mL (Figure 3). Conversely, when Hsf5 and Hsf6 were combined, they exhibited 63% and 98% trichomonacidal activity, respectively, at the highest evaluated concentrations. The effect of MTZ in combination with Hsf5 and Hsf6 did not potentiate their activity, as indicated by the IC_50_ values of 64.28 and 106.9 µg/mL, respectively. These values were calculated using nonlinear regression by plotting the logarithm of concentration with the GraphPad Prism^®^ software.

### 3.4. GC/MS Analysis of Hsf6

Hsf6 was analyzed through GC-MS owing to its substantial trichomonacidal activity and low cytotoxicity. The mass spectrum of the analyzed sample revealed three prominent peaks corresponding to the major compounds and thirteen peaks below 6%, with these three peaks representing 74.572% of the total sample (Figure 4). The main compounds identified in Hsf6 were pregn-16-en-20-one, 3-hydroxy-, [3β,5β]-; bis(2-ethylhexyl) phthalate; and 3-(1,5-dimethyl-hexyl)-3a,10,10,12b-tetramethyl-1,2,3,3a,4,6,8,9,10,10a,11,12,12a,12b-tetradecahydro-benzo[4,5]cyclohepta[1,2-E]indene, with percentages of 44.93%, 17.97%, and 11.66%, respectively (Table 4). These identified compounds can be grouped into three main categories: alkanes such as hexadecane; alkenes such as eicosane (which do not show relevant biological activity reports); and finally, the group of terpenoids which encompasses most of the molecules identified in this analysis. Among them, the sesquiterpene α-cadinol stands out, along with molecules reported to have antimicrobial, repellent, and antiparasitic activities identified in other medicinal plants.

Six compounds identified in the Hsf6 sample were determined to be aliphatic chains, including alkanes and alkenes. Collectively, these compounds constituted 6.951% of the composition of Hsf6. Additionally, other molecules identified as terpenes (which are found in natural products and frequently exhibit biological activities) were found. Among the identified terpenes, the sesquiterpene α-cadinol stands out, exhibiting antihypertensive activity. In addition, terpenes reported in extracts of medicinal plants with antibacterial activity, as 2-pentadecanone-6, 10, 14-trimethyl-, 7-dehydrodiosgenin, and pregn-16-en-20-one, 3-hydroxy-, [3β,5β], were identified. These compounds may be of interest for further pharmacological research given their antibacterial properties (Figure 5) [28,29,30,31,32]. In addition, stigmasta-3,5-dien-7-one, a terpene with reports of antiparasitic activity was detected [32]. Other antibacterial molecules and membrane stabilizers such as dibutyl phthalate and β-sitosterol acetate, were detected in the analyzed subfraction [33,34].

## 4. Discussion

*P. crassinervata* has a documented history in Querétaro, Mexico, where an infusion from its fronds is employed to disinfect mouth ulcers [6]. These injuries can have multifactorial origins including bacterial accumulation or tongue flaking, commonly observed in patients with acute gastroenteritis caused by diverse pathogens such as *S. typhi* and *S. flexneri* infections [35].

Previous research on *P. crassinervata* has primarily focused on ecological and taxonomic aspects, encompassing morphological and phylogenetic descriptions [36,37,38,39]. In previous studies, the MICs of 505 extracts obtained from 185 species (75 families) were evaluated against a methicillin-resistant *S. aureus strain* (known for its resistance to beta-lactam antibiotics) with an MIC value < 128 μg/mL [40]. In our study, precipitated fraction MIC values averaged 125 μg/mL. This value is comparable to those of natural products evaluated by other researchers. This validates its traditional use as an antimicrobial agent. Molecules with dual activity, functioning as both antibacterial and antioxidant agents, have been reported in natural products. Such compounds hold promise for the development of more effective therapeutic solutions, particularly in addressing complex and multigenic diseases. For instance, eugenol, a polyphenol, exhibits dual functionality as an antioxidant and antibacterial agent. It has been shown to be effective against both Gram-positive and Gram-negative bacteria with MICs as low as 2.5 µg/mL [41]. We observed that the water-soluble fraction of *P. crassinervata* demonstrated the most robust activity in both assessments. According to previous investigations conducted by our research group, this fraction contains approximately 14.38 ± 0.33% of polyphenols, which may be contributing toward the dual activities, as antioxidants and antimicrobial agents [19].

Additionally, in the evaluation against *T. vaginalis*, Hf exhibited the highest activity of all the natural products analyzed. It is noteworthy that several researchers consider IC_50_ values ≤ 100 μg/mL as indicative of significant antiparasitic activity (Ibañez-Escribano, 2015). Trichomoniasis is responsible for more than half of the annual cases of curable STIs [15]. MTZ is the primary drug used to treat trichomoniasis. However, it is associated with significant drawbacks including high toxicity and genotoxicity, interactions with steroids and antipyretics, and near 10% of clinical isolates are MTZ-resistant [42,43].

Consequently, there is an urgent need to develop new and effective trichomonacidal treatments. In this context, the WHO has advocated the exploration of natural products as a promising source for potential trichomonacidal agents [44,45]. Recently, there has been a growing interest in natural products in the search for novel drugs in the field of parasitology, particularly in the fight against *Trichomonas vaginalis*. Our results, particularly with Hf, indicate an interesting level of activity against the parasite. This approach supports and substantiates our research as the findings obtained support the notion that medicinal plants harbor compounds with significant pharmacological activities. These results underline the importance of exploring and harnessing the wealth of natural resources in the search for effective therapies against parasitic diseases [46].

Recent investigations into *P. crassinervata* have shed light on its chemical components, revealing the presence of terpenoid-like molecules, tannins, and alkaloids in qualitative analyses [19]. These compounds are known for their diverse range of biological activities. Notably, terpenoid compounds such as phytol have been identified in the Hsf1 subfraction [20]. Additionally, an investigation revealed that Hf from *P. crassinervata* exerted an effect on *T. gondii* tachyzoites with an IC_50_ value as low as 16.9 µg/mL. This finding is particularly noteworthy given the taxonomic similarities between *T. vaginalis* and *T. gondii*, as both protozoan parasites are affected by the same fractions [19,20]. This study showed that Hsf6 with trichomonacidal activity is primarily composed of terpenes. Considering that terpenoids have demonstrated activity against protozoal species like *T. vaginalis* and *Leishmania infantum*, and terpenoids have been identified in Hf, it is plausible that these compounds contribute to the observed activity [6,22]. Notably, the CC_50_ for MTZ in Vero cells is considerably higher at 70 µg/mL [23].

Consequently, the trichomonacidal IC_50_ values obtained with the Hf of *P. crassinervata* appear to fall within a promising range. Terpenes can exert their effects through various mechanisms, including interference with generating reactive oxygen species, the disruption of genetic material replication, and the modulation of calcium transporters like artemisinin, parthenin, and piperazine [47]. In addition, terpenes can significantly impact essential targets, inhibiting metabolic pathways such as fatty acid synthesis, parasitic proteases, and even membrane disruption [32].

Nonetheless, further studies are warranted to comprehensively assess Hf using in vitro and in vivo models and elucidate its potential action mechanism. This study represents the first demonstration of the efficacy of Hf against both Gram-positive and Gram-negative bacteria and *T. vaginalis*. This prompts further exploration to identify the active compounds in Hsf5 and Hsf6. Notably, a synergistic effect can be ruled out as their effects are additive. However, it is plausible that other subfractions may be contributing to enhancing the effect against the parasite as the activity is higher in the fraction before the separation of subfractions. This complexity arises from the diversity of mechanisms exhibited by terpenes. This suggests a complex interplay of compounds within *P. crassinervata* that warrants further investigation. Subsequent studies are required to identify the active compounds and conduct complementary research to elucidate the underlying mechanism of action.

In the present study, various promising phytoconstituents were found to be effective against both Gram-positive and Gram-negative bacteria and *T. vaginalis*. Resistance to antimicrobial drugs is a challenge of global medical importance. Its consequences are intensified by poverty and inequality, mostly in low- and middle-income countries. Despite efforts to overcome antimicrobial drug resistance and toxicity, there have been no satisfactory results so far. Natural products could provide a vast pool of compounds to address these problems.

## 5. Conclusions

In conclusion, *P. crassinervata* exhibited activity against both Gram-positive and Gram-negative bacteria and it was also effective against *T. vaginalis*, with relevant activity values and no observed cytotoxic effects, even at concentrations of up to 200 µg/mL. Our study underscores the potential of *P. crassinervata* as a source of bioactive compounds with its dual functionality as a trichomonacidal and antibacterial agent highlighting its ability to address complex diseases. This prompts further research to unravel mechanisms of action from active compounds. Overall, *P. crassinervata* emerges as a promising candidate for effective treatments for gastrointestinal infections and trichomoniasis. This research substantiates the use of *P. crassinervata* in traditional Mexican medicine and emphasizes its potential as a source of bioactive compounds. Additional research and clinical studies may further establish its role in drug discovery.

## Figures and Tables

**Figure 1 pharmaceutics-16-00624-f001:**
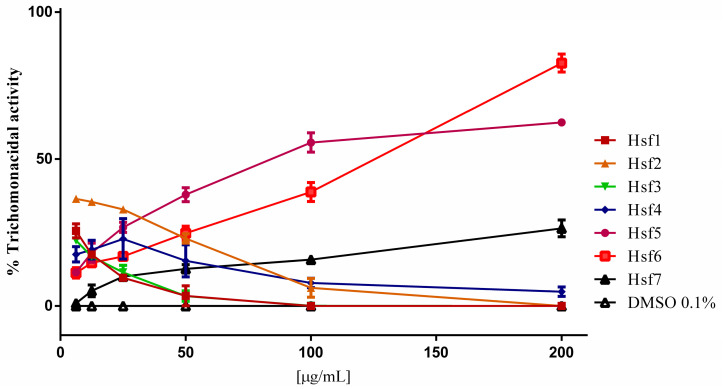
*T. vaginalis* trophozoites after 24 h exposure to *P. crassinervata* subfractions. Negative control: DMSO at a maximum concentration of 0.1%.

**Figure 2 pharmaceutics-16-00624-f002:**
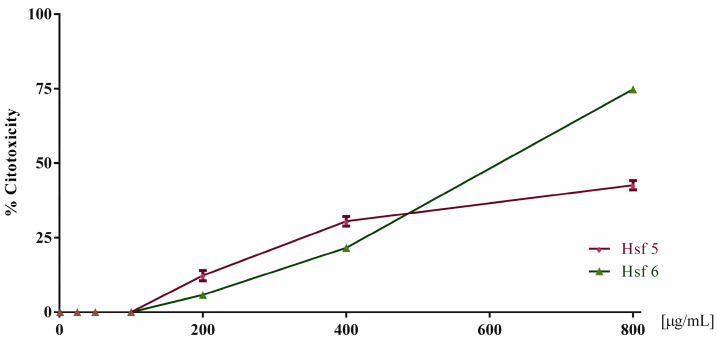
Toxicity percentage in Vero cell cultures exposed to Hsf5 and Hsf6 for 24 h at a concentration range of 25–800 µg/mL for growth control. A control with 0.1% DMSO was included, showing 0% toxicity.

**Figure 3 pharmaceutics-16-00624-f003:**
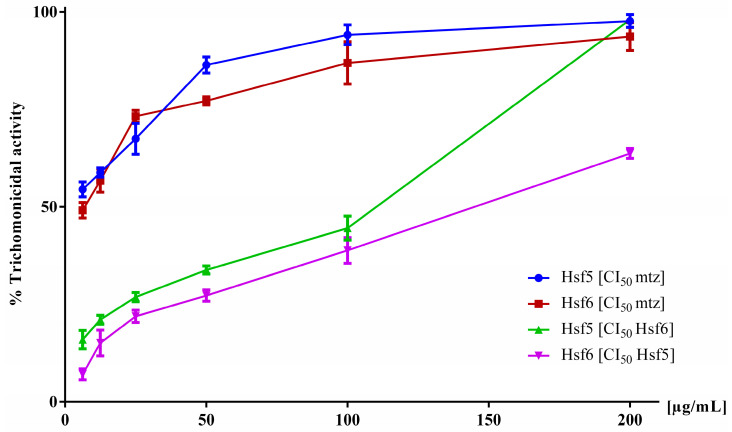
In vitro trichomonacidal activity percentage in the axenic cultures of *T. vaginalis* exposed to mixtures of Hsf5 and Hsf6 for 24 h while maintaining the IC_50_ of one Hsf constant in each assessment. Growth control (no-treatment) and 0.1% DMSO control were included, exhibiting 0% activity.

**Figure 4 pharmaceutics-16-00624-f004:**
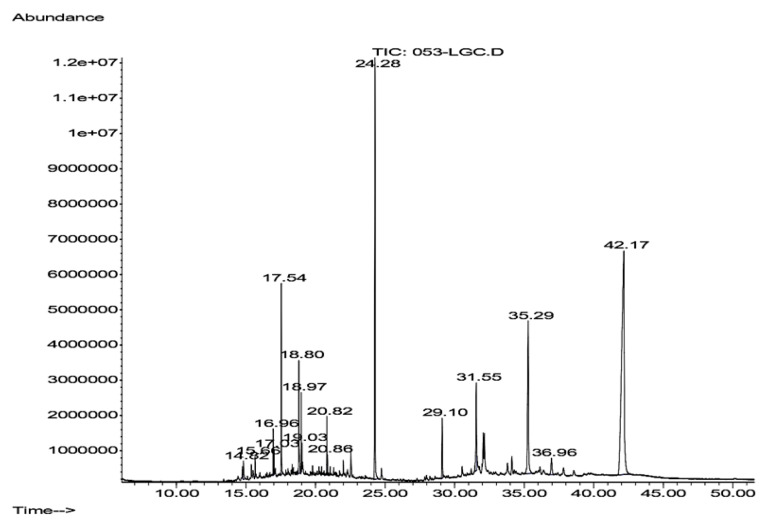
Chromatogram of Hsf6 obtained after gas–mass chromatography (GC-MS) analysis for 1 h. Each peak corresponds to the spectrum of each compound; 15 peaks were detected.

**Figure 5 pharmaceutics-16-00624-f005:**
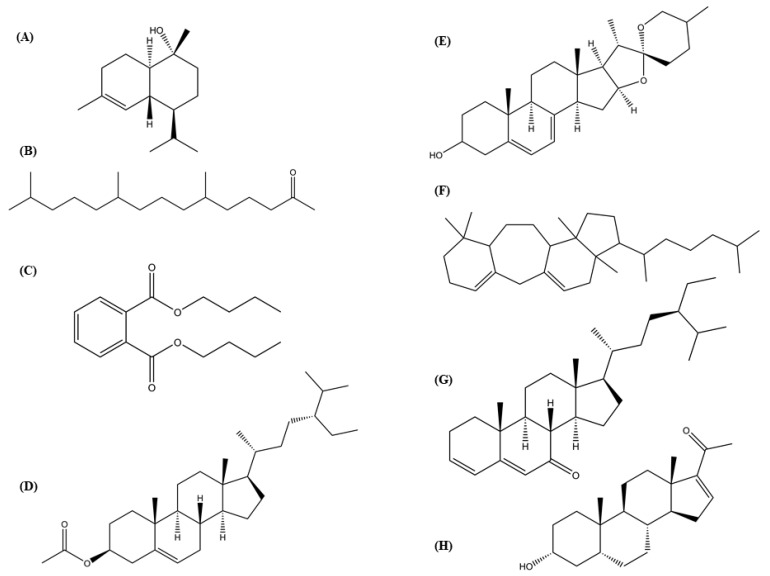
Chemical structures of eight compounds from the hexane subfraction 6 (Hsf6) with biological activity reported. (**A**) α-Cadinol; (**B**) 2-pentadecanone-6,10,14-trimethyl-; (**C**) dibutylphthalate; (**D**) β-sitosterol acetate; (**E**) 7-dehydrodiosgenin; (**F**) 3-(1,5-dimethyl-hexyl)-3a,10,10,12b-tetramethyl-1,2,3,3a,4,6,8,9,10,10a,11,12,12a,12b-tetradecahydro-benzo[4,5]cyclohepta[1,2-E]indene; (**G**) stigmasta-3,5-dien-7-one; (**H**) pregn-16-en-20-one, 3-hydroxy-, [3β,5β]. Structures of alkanes and alkenes without reported biological activity were excluded.

**Table 1 pharmaceutics-16-00624-t001:** Minimum inhibitory concentration (µg/mL) of extracts from *Pleopeltis crassinervata* organs against pathogenic bacteria.

Organ	Extract	Bacterial Strain						
		A	B	C	D	E	F	G
		MIC (μg/mL)						
Frond	Hexane	>1500	>1500	>1500	>1500	>1500	>1500	>1500
	Methanol	>1500	747	373	1500	1500	>1500	>1500
	Aqueous	>1500	1500	>1500	>1500	1500	>1500	>1500
Rizome	Hexane	>1500	>1500	>1500	>1500	>1500	>1500	>1500
	Methanol	>1500	1500	377	>1500	1500	>1500	>1500
	Aqueous	>1500	1500	>1500	>1500	1500	>1500	>1500
Root	Hexane	ND	ND	ND	ND	ND	ND	ND
	Metanol	>1500	747	>1500	1500	1500	>1500	>1500
	Aqueous	>1500	1500	>1500	>1500	1500	>1500	>1500

Bacterial strains: (A) *Salmonella typhimurium*, (B) *Salmonella typhi* ATCC 6539, (C) *Staphylococcus aureus* ATCC 6538, (D) *Proteus mirabilis* NTC 289, (E) *Shigella flexneri* ATCC 29003, (F) *Bacillus subtilis*, (G) *Escherichia coli* ATCC8739. ND: Not determined.

**Table 2 pharmaceutics-16-00624-t002:** The minimum inhibitory concentration (MIC) (µg/mL) of *Pleopeltis crassinervata* frond methanolic extract fractions.

Bacteria	MeOH Extract	Hexanic Fraction	Precipitate	MetOH Fraction
	MIC (µg/mL)			
*Escherichia colli*	>1500	>1500	>1500	>1500
*Salmonella typhimurium*	1000	1000	1000	1000
*Salmonella typhi*	500	1000	500	500
*Shigella flexneri*	1000	1000	500	500
*Staphylococcus aureus*	250	1000	125	>1500
*Bacillus subtilis*	>1500	>1500	>1500	>1500
*Proteus mirabilis*	>1500	>1500	1000	>1500

MetOH: methanolic, MIC: minimum inhibitory concentration.

**Table 3 pharmaceutics-16-00624-t003:** Trichomonacidal activity by *Pleopeltis crassinervata* frond methanolic extract, and its fractions.

Sample	CI_50_ [µg/mL]
Frond methanolic extract	>200
Methanolic fraction	>200
Hexane fraction	82.3
Precipitate fraction	>200
MTZ	<4

**Table 4 pharmaceutics-16-00624-t004:** Compounds identified in the GC-MS analysis of Hsf6 via spectra correlation using the NIST database 17.a.

Elution Order	Compound	Retention Time (min)	Formula	Area %	Synonyms
1	Hexadecane	14.821	C_16_H_34_	0.813	n-Cetane
2	α-Cadinol	15.662	C_15_H_26_O	0.607	Cadin-4-en-10-ol
3	1-Octadecene	16.963	C_18_H_36_	3.050	Octadec-1-ene
4	Octadacane	17.028	C18H38	0.668	n-Octadecane
5	2-Pentadecanone-6,10,14-trimethyl-	17.541	C_18_H_36_O	5.170	Fitone
6	(M)Dibutylphthalate	18.802	C_16_H_22_O_4_	4.046	dibutyl benzene-1,2-dicarboxylate
7	Eicosane	19.032	C_20_H_42_	0.577	Icosane
8	Docosene	20.819	C_22_H_44_	1.370	docos-1-ene
9	Docosane	20.858	C_22_H_46_	0.473	n-Docosane
10	Bis(2-ethylhexil) phthalate	24.280	C_24_H_38_O_4_	17.975	Di(2-ethylhexyl)phthalate
11	β-Sitosterol acetate	29.102	C_31_H_52_O_2_	2.761	Acetyl-beta-sitosterol
12	7-Dehydrodiosgenin	31.552	C_27_H_40_O_3_	4.422	Spirosta-5,7-dien-3-ol
13	3-(1,5-Dimethyl-hexyl)-3a,10,10,12b-tetramethyl-1,2,3,3a,4,6,8,9,10,10a,11,12,12a,12b-tetradecahydro-benzo[4,5]cyclohepta[1,2-E]indene	35.283	C_30_H_50_	11.663	1(10),9(11)-B-Homolanistadiene
14	Stigmasta-3,5-dien-7-one	36.964	C_29_H_46_O	1.471	Tremulone
15	Pregn-16-en-20-one, 3-hydroxy-, [3β,5β]-	42.167	C_21_H_32_O_2_	44.934	16-Pregnenolone

## Data Availability

Data are contained within the article.

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
