# Peer review of "Antibacterial, Trichomonacidal, and Cytotoxic Activities of Pleopeltis crassinervata Extracts"

_pharmaceutics, 2024, doi:10.3390/pharmaceutics16050624_

Round 1

Reviewer 1 Report

Comments and Suggestions for Authors

Antibacterial, trichomonacidal, and cytotoxic activities of Pleopeltis crassinervata extracts

 Authors investigated the biological activity of extracts and fractions obtained from Pleopeltis crassinervata (Pc) organs against bacteria and Trichomonas vaginalis using in vitro models.

It is an interesting work. In the context of the antibiotic resistance crisis that the world is facing, it is important to find alternatives to these antibiotics, and the authors show some promising results. It becomes even more interesting as authors study products already used in the community, in traditional medicine.

 There are some points that need revision:

 -          “bacteria were cultivated”. Cultivation of bacteria is not the correct way of mention the growth of microorganisms. It must be corrected to “were inoculated” or “growth in…”. All document should be revised.

-          Materials and Methods, 2.1., line 111:

Authors used 5% CO2 to growth bacteria that doesn’t need it. Why? All bacteria mentioned growth well in aerobiosis.

       -          Materials and Methods, 2.5., line 157:

penicillin/streptomycin (PenStrep) mixture is a positive control, right? DMSO and PSS will have different results from PenStrep, first two don’t have antimicrobial action, the PenStrep has it. So, first ones are Negative controls, the PenStrep is a positive control.

      -           Materials and Methods, 2.7., line 188:

The same with Metronidazol as stated with PenStrep. Metronidazol is a positive control, DMSO and RPMI are negative controls.

 -          Conclusion

The statement “Pc exhibited significant activity against both gram-positive and gram-negative bacteria, as well as T. vaginalis” should be replaced by “Pc exhibited activity against both gram-positive and gram- negative bacteria, as well as T. vaginalis”. It is not true that is a significant activity, there is some activity, only against some bacteria, and the MICs obtained are not significant.

 -          6. Patents ad Supplementary materials

Nothing is mentioned in both chapters, so it seems that these two should not be present!

     -          Authors must mention/explain why they did not carry out tests to determine MICs in accordance with the indications of The Clinical & Laboratory Standards Institute (CLSI) or The European Committee on Antimicrobial Susceptibility Testing

Comments on the Quality of English Language

A revision of all document must take place. 

Author Response

(Reviewer 1)

Antibacterial, trichomonacidal, and cytotoxic activities of Pleopeltis crassinervata extracts

 Authors investigated the biological activity of extracts and fractions obtained from Pleopeltis crassinervata (Pc) organs against bacteria and Trichomonas vaginalis using in vitro models.

It is an interesting work. In the context of the antibiotic resistance crisis that the world is facing, it is important to find alternatives to these antibiotics, and the authors show some promising results. It becomes even more interesting as authors study products already used in the community, in traditional medicine.

There are some points that need revision:

 -          “bacteria were cultivated”. Cultivation of bacteria is not the correct way of mention the growth of microorganisms. It must be corrected to “were inoculated” or “growth in…”. All document should be revised. The modification was made to all document.

Response

Changes were made to the document (highlighted in yellow).

-          Materials and Methods, 2.1., line 111:

Authors used 5% CO2 to growth bacteria that doesn’t need it. Why? All bacteria mentioned growth well in aerobiosis.

Response

There was an error in the procedure: CO2 atmosphere was not used in the bacterial inoculum; they were solely incubated at 37 °C. The correction has been made in the text.

       -          Materials and Methods, 2.5., line 157:

penicillin/streptomycin (PenStrep) mixture is a positive control, right? DMSO and PSS will have different results from PenStrep, first two don’t have antimicrobial action, the PenStrep has it. So, first ones are Negative controls, the PenStrep is a positive control.

Response

Thank you for bringing this to our attention. We have addressed the error in the text as you suggested.

      -           Materials and Methods, 2.7., line 188:

The same with Metronidazol as stated with PenStrep. Metronidazol is a positive control, DMSO and RPMI are negative controls.

Response:

Metronidazole is the positive control for Trichomonas vaginalis assays, as it is the approved reference drug for the treatment of trichomoniasis. However, it does not have a non-specific cytotoxic effect on mammalian cells, so we included it in the Vero CCL-81 assays, to calculate selectivity index. However, with the reviewer comment we realized that we were not clear enough in this section, and have changed the paragraph to read as follows:

Line 193: “DMSO in RPMI (0.1% v/v) was used as negative control

 -          Conclusion

The statement “Pc exhibited significant activity against both gram-positive and gram-negative bacteria, as well as T. vaginalis” should be replaced by “Pc exhibited activity against both gram-positive and gram- negative bacteria, as well as T. vaginalis”. It is not true that is a significant activity, there is some activity, only against some bacteria, and the MICs obtained are not significant.

Response:

The statement has been revised based on your suggestion.

 -          6. Patents ad Supplementary materials

Nothing is mentioned in both chapters, so it seems that these two should not be present!

Response:

Thanks for noticing, it was a mistake and it has already been corrected

     -          Authors must mention/explain why they did not carry out tests to determine MICs in accordance with the indications of The Clinical & Laboratory Standards Institute (CLSI) or The European Committee on Antimicrobial Susceptibility Testing.

Response:

Dear reviewer, The techniques standardized by the CLSI and EUCAST are used in drugs and combinations of them. However, when natural products are tested, it is necessary to make modifications to the standardized protocols. We decided to use the resazurin assay since it is a versatile test that allows the detection of microbial growth in small volumes of solution in microplates. The microdilution rationale is similar to that recommended by CLSI. This rezasurin test measures the metabolic activity of living cells. that color changes from blue (resazurin) to pink (resorufin) can continue to be monitored, demonstrating a clear indication of the presence or absence of viable microorganisms. This assay also allows quantification of the activity of antimicrobial properties by determining the MIC. On the other hand, this method is standardized, precise, economical, and easy to perform and has been widely used in the evaluation of natural products.

Balouiri, M., Sadiki, M., & Ibnsouda, S. K. (2016). Methods for in vitro evaluating antimicrobial activity: A review. Journal of pharmaceutical analysis, 6(2), 71-79.

Comments on the Quality of English Language

A revision of all document must take place. 

Response:

The manuscript underwent revision through Elsevier Language Editing Services. Order reference: ASLESTD1046153

Reviewer 2 Report

Comments and Suggestions for Authors

This article presents some data for extraction and biological activities study of active substances from Pleopeltis crassinervata. The work has good scientific impact. However, I have many major remarks to the authors:

1. Authors need to know some general rules for writing the manuscript. For example: DO NOT make abbreviations in the Abstract section. They have to be defined in the first place in the text where they are used. When define some abbreviation use it further in the manuscript do not use the whole term again. What does RPMI mean? Use always a passive voice in the article. These are not true for this manuscript.

2. I found that the Introduction section is not an introduction in the studied problem but it is written to show how great is Mexico in the world. It has to be clearly directed to the studied problem.

3.  Why all numbers are presented with a comma in the number. For example, is 21 989 true or 21,989 is true. The meaning of both numbers is absolutely different

4. Description of the Material and methods part and results is absolutely disaster.

-          Line 131 – the organs were subjected to extraction for 4-5 days. There are no any conditions described for the process. The same in line 134-135.

-          The lyophilized powder is dissolved in methanol at 10% concentration (line 137). And what is the rest 90%

-          Line 138 – three fractions are obtained (hexane, methanol and precipitate). Is it a precipitate or it is insoluble in both solvent fraction?

-          Why authors use different concentrations for Antibacterial assays and Trichomonacidal assay? Do they have data from literature? If yes, it is not clear how they choose the work concentrations.

6.      Line 206 Hsf6 was isolated through column chromatography. What are the conditions, solid and mobile phases, etc?

7.      Line 227 repeat line 145

8.      It is not clear the made extracts are lyophilized and further dissolved in DMSO for the study or how?

9.      Authors described in the text that they work in the MIC of extracts but finally they report many NA data in table 1 as NOT active in a concentration less than 1.5 mg/mL. So, if they work at MIC why they do not report MIC in the table but say that the extracts have no activity in a low concentration?

     5. Table 3 is absolutely not well presented. There is activity and below concentration and numbers. So, which number below is activity and which one is concentration? Moreover, if 100 is activity, what they mean as activity 100%? Is this 100% inhibition or 100% growth?

16. Where is Figure 6 cited n line 360?

17. If authors did not study the hexane extract activity due to low solubility in DMSO (line 228) why further authors present to the readers’ attention the content of the extract (Figure 5) studied by GC-MS?

18. I found that many of the discussion is presented in the Results section and the Discussion one is absolutely not written in a good manner. It repeats the results.

Finally, my opinion is that this article has good scientific impact, but it is not presented clearly and in a good manner.  

Comments on the Quality of English Language

English language of the paper is very difficult to be red. There are many absolutely incomprehensible sentences. Moreover, authors use long sentences, for example lines 93-97 (5 lines), which make this sentence absolutely not clear for readers.

Author Response

(Reviewer 2)

Response

Changes were made to the document (highlighted in yellow).

This article presents some data for extraction and biological activities study of active substances from Pleopeltis crassinervata. The work has good scientific impact. However, I have many major remarks to the authors:

  1. Authors need to know some general rules for writing the manuscript. For example: DO NOT make abbreviations in the Abstract section. They have to be defined in the first place in the text where they are used. When define some abbreviation use it further in the manuscript do not use the whole term again. What does RPMI mean? Use always a passive voice in the article. These are not true for this manuscript.

Response:

We would like to express our sincere gratitude for taking the time and effort to review our manuscript. Thank you for your feedback. We have considered your comments and made the necessary corrections in the text.

  1. I found that the Introduction section is not an introduction in the studied problem but it is written to show how great is Mexico in the world. It has to be clearly directed to the studied problem.

Response:

In reference to your observation about introduction, we consider that the inclusion of information on the diversity of medicinal plants in Mexico is relevant to provide the reader with an understanding of the context in which our research is developed. This study arises as a bioguided research in search of the biological activities of the plant, its safety and phytochemical composition. The plant object of this study belongs to a group poorly studied in the field of pharmacognosy. We think that it is important to highlight the fact that, despite the diversity of medicinal plants in our country, empirical knowledge is being lost and how these could contain molecules of therapeutic interest. Its evaluation is crucial, especially considering that a large part of the population uses natural products to treat basic health problems, and studies of this type validate or refute their use.

  1. Why all numbers are presented with a comma in the number. For example, is 21 989 true or 21,989 is true. The meaning of both numbers is absolutely different

Response:

The use of commas differs in the Spanish language, and we recognize that we made a mistake when writing in English. We appreciate you pointing it out; We have corrected this error throughout the text.

  1. Description of the Material and methods part and results is absolutely disaster.

-          Line 131 – the organs were subjected to extraction for 4-5 days. There are no any conditions described for the process. The same in line 134-135.

Response:

The extraction conditions were added.

-          The lyophilized powder is dissolved in methanol at 10% concentration (line 137). And what is the rest 90% 

Response:

Our intention was not clear when we wrote that sentence. We wanted to convey that the lyophilized powder was dissolved at a ratio of 1:10 W/V. The sentence has been corrected.

-          Line 138 – three fractions are obtained (hexane, methanol and precipitate). Is it a precipitate or it is insoluble in both solvent fraction?

Response:

Only two solvents, methanol and hexane, were used to obtain the fractions. Any compound that did not dissolve precipitated out, and this fraction was named as the precipitate, which was soluble in water.

-          Why authors use different concentrations for Antibacterial assays and Trichomonacidal assay? Do they have data from literature? If yes, it is not clear how they choose the work concentrations.

Response:

The evaluated concentrations were determined based on the solubility of the extracts and in twofold dilutions from the highest evaluated concentration. This allowed for the assessment of different concentrations and thus the calculation of MIC or IC50 values in bacteria and T. vaginalis, respectively. The limitation in the concentrations assayed has been the solubility of each extract. According to T. vaginalis assays, different researchers have used for trichomonacidal assays concentrations between 500 µg/ml and 100 µg/ml. Following the criteria used by Nogueira Barbosa et al. 2023, Tawfeek and Sarhan 2019 or Martínez-Díaz et al., 2015, we have performed the assays against the parasite using 200 µg/ml as the highest concentration. Given the question raised by this reviewer, we felt it was appropriate to include these references.

  1. Line 206 Hsf6 was isolated through column chromatography. What are the conditions, solid and mobile phases, etc?

Response: In this section we do not give more details about the process because our research group has published it. We have cited this information from our previous post to avoid confusion in the text.

  1. Line 227 repeat line 145

Response:The repeated sentence on line 227 was removed.

  1. It is not clear the made extracts are lyophilized and further dissolved in DMSO for the study or how?

Response:

The section has been rewritten for better understanding. All extracts were lyophilized to remove moisture and accurately express the mass in each evaluation. DMSO was used as a vehicle to prepare stock solutions.

  1. Authors described in the text that they work in the MIC of extracts but finally they report many NA data in table 1 as NOT active in a concentration less than 1.5 mg/mL. So, if they work at MIC why they do not report MIC in the table but say that the extracts have no activity in a low concentration?

Response:

The maximum evaluated concentration was 1.5 mg/mL. This table presents all extracts obtained, three from each organ, and evaluated against seven bacterial strains. Only the MIC values of active extracts are shown; "N/A" indicates they were inactive at those concentrations, and "ND" is added to extracts not evaluated due to their yield or solubility.

  1. Table 3 is absolutely not well presented. There is activity and below concentration and numbers. So, which number below is activity and which one is concentration? Moreover, if 100 is activity, what they mean as activity 100%? Is this 100% inhibition or 100% growth?

Response:

We have reconsidered the presentation of the data in the table, as we noted that it could cause confusion and the data was not optimally arranged. We decided to present the data in a different way, highlighting IC50 values, which are important pharmacological parameters. Detailed results are now described in the text to provide a clearer understanding. We hope that this new presentation will facilitate the interpretation of the results

  1. Where is Figure 6 cited n line 360?

Response:It was our mistake, Figure 6 does not exist. It has been removed from the text.

  1. If authors did not study the hexane extract activity due to low solubility in DMSO (line 228) why further authors present to the readers’ attention the content of the extract (Figure 5) studied by GC-MS?

Response:

We think there might have been some misunderstanding regarding the quantity of extracts assessed. It's important to differentiate between the hexane extract and the hexane fraction. Specifically, the text notes that the extract with low solubility was the hexane extract from the roots. However, the hexane fraction, obtained from the methanolic extract of the frond, was indeed subjected to evaluation.

  1. I found that many of the discussion is presented in the Results section and the Discussion one is absolutely not written in a good manner. It repeats the results.

Response:

We have reviewed the wording and corrected it according to your observations so as not to repeat data in the discussion that were described in results.

Finally, my opinion is that this article has good scientific impact, but it is not presented clearly and in a good manner.  

Comments on the Quality of English Language

English language of the paper is very difficult to be red. There are many absolutely incomprehensible sentences. Moreover, authors use long sentences, for example lines 93-97 (5 lines), which make this sentence absolutely not clear for readers.

Response: We considered their perspective to improve the manuscript, which was reviewed by Elsevier Language Editing Services. Order reference: ASLESTD1046153

Reviewer 3 Report

Comments and Suggestions for Authors

The manuscript presents data on antibacterial, trichomonacidal, and cytotoxic activities of Pleopeltis crassinervata extracts. It can be reconsidered for publication after critical analysis of experimental data and implementation of certain information on materials and methods used.

Below there are some specific comments/suggestions for its improvement.

Authors should explain why they used resazurin method instead of standard methods of antimicrobial susceptibility testing (e.g. CLSI, 2009; FDA, 2018; ISO 2019).

The results of antimicrobial assay should be expressed in μg/ml.

In case of the results of antimicrobial assay, minimum inhibitory concentration (MIC) values below 100 μg/ml for extracts should be considered as promising activity/highly effective. In addition, samples with MICs higher than 1000 μg/ml should strictly be evaluated as no active (Kokoska et al., 2019). Therefore, the interpretation of results achieved for the extracts tested in the study should be re-evaluated according to the above-mentioned criteria in the text and Table 1 (in general, the results are showing weak or no activity).

When Latin name of any organism is mentioned in the text, its abbreviated versions should be used in rest of the manuscript (e.g. lines 63 and 126: P. crassinervata).

Line 157: antibiotics should be tested as “positive” controls.

The identification of constituents using GC/MS analysis is based on comparison of mass spectra and retention times only. Calculation of retention indices and coelution with authentic standards (in case of certain compounds) is necessary (Table 4).

References

CLSI 2009. Methods for Dilution Antimicrobial Susceptibility Tests for Bacteria That Grow Aerobically; Approved Standard-Eighth Edition. CLSI document M07-A8. Wayne, PA, USA: Clinical Laboratory Standards Institute.

FDA Guidance 2018. Class II Special Controls Guidance Document: Antimicrobial Susceptibility Test Systems.

ISO 2019. Susceptibility Testing of Infectious Agents and Evaluation of Performance of Antimicrobial Susceptibility Devices, Part 1. Broth micro-dilution reference method for testing the in vitro activity of antimicrobial agents against rapidly growing aerobic bacteria involved in infectious diseases. Second edition. ISO/DIS 20776-1, Geneva, Switzerland.

Kokoska L, Kloucek P, Leuner O, Novy P. 2019. Plant-derived products as antibacterial and antifungal agents in human health care. Current Medicinal Chemistry, 26(29): 5501-5541.

Comments on the Quality of English Language

Moderate editing of English language required.

Author Response

Reviewer 3)

Response

Changes were made to the document (highlighted in yellow).

The manuscript presents data on antibacterial, trichomonacidal, and cytotoxic activities of Pleopeltis crassinervata extracts. It can be reconsidered for publication after critical analysis of experimental data and implementation of certain information on materials and methods used.

Response:

 We would like to express our sincere gratitude for taking the time and effort to review our manuscript. We appreciate your comments. We have considered your comments and made the necessary corrections to the text.

Below there are some specific comments/suggestions for its improvement.

Authors should explain why they used resazurin method instead of standard methods of antimicrobial susceptibility testing (e.g. CLSI, 2009; FDA, 2018; ISO 2019).

Response: Dear reviewer, The techniques standardized by the CLSI and EUCAST are used in drugs and combinations of them. However, when natural products are tested, it is necessary to make modifications to the standardized protocols. We decided to use the resazurin assay since it is a versatile test that allows the detection of microbial growth in small volumes of solution in microplates. The microdilution rationale is similar to that recommended by CLSI. This rezasurin test measures the metabolic activity of living cells. that color changes from blue (resazurin) to pink (resorufin) can continue to be monitored, demonstrating a clear indication of the presence or absence of viable microorganisms. This assay also allows quantification of the activity of antimicrobial properties by determining the MIC. On the other hand, this method is standardized, precise, economical and easy to perform and has been widely used in the evaluation of natural products. 

Balouiri, M., Sadiki, M., & Ibnsouda, S. K. (2016). Methods for in vitro evaluating antimicrobial activity: A review. Journal of pharmaceutical analysis, 6(2), 71-79.

The results of antimicrobial assay should be expressed in μg/ml.

Response: We agree that it is more appropriate to express the MIC values in µg/mL. These values were converted from mg/mL to µg/mL in the antibacterial assays.

In case of the results of antimicrobial assay, minimum inhibitory concentration (MIC) values below 100 μg/ml for extracts should be considered as promising activity/highly effective. In addition, samples with MICs higher than 1000 μg/ml should strictly be evaluated as no active (Kokoska et al., 2019). Therefore, the interpretation of results achieved for the extracts tested in the study should be re-evaluated according to the above-mentioned criteria in the text and Table 1 (in general, the results are showing weak or no activity).

Response: We have reviewed the literature provided and agree that the extracts do not exhibit antibacterial activity considered highly effective. However, we consider that the MIC values obtained are relevant for their presentation. It is common in publications on natural products to report MIC values even above 1000 µg/mL. This section was corrected according to your suggestion and in our results, we highlight that the MIC values that are considered relevant are those below 100 µg/mL, which indicates a low activity of our extracts.

When Latin name of any organism is mentioned in the text, its abbreviated versions should be used in rest of the manuscript (e.g. lines 63 and 126: P. crassinervata).

Response: It was reviewed all scientific names are spelled correctly throughout the document, and any errors were corrected.

Line 157: antibiotics should be tested as “positive” controls.

Response: We agree with your viewpoints; antibiotics serve as positive controls. The correction has been made.

The identification of constituents using GC/MS analysis is based on comparison of mass spectra and retention times only. Calculation of retention indices and coelution with authentic standards (in case of certain compounds) is necessary (Table 4).

Response: CG/MS analysis is highly sensitive and has been validated through decades of research and development in the scientific community. It can detect even trace amounts of compounds in a sample. In addition, mass spectrometry provides high selectivity when identifying compounds based on their unique fragmentation patterns. In many publications only this analysis is used to identify compounds present in studies related to natural products. We appreciate your comments and agree that some compounds could be corroborated by standards, however, not all are found commercially and it would be necessary to resort to chemical synthesis. At this point, it would not be feasible to perform these experiments and include them in this document. We plan to perform future biological evaluations with the identified compounds, and these standards could be included but we consider that this manuscript contains the assays necessary for a publication.

References

CLSI 2009. Methods for Dilution Antimicrobial Susceptibility Tests for Bacteria That Grow Aerobically; Approved Standard-Eighth Edition. CLSI document M07-A8. Wayne, PA, USA: Clinical Laboratory Standards Institute.

FDA Guidance 2018. Class II Special Controls Guidance Document: Antimicrobial Susceptibility Test Systems.

ISO 2019. Susceptibility Testing of Infectious Agents and Evaluation of Performance of Antimicrobial Susceptibility Devices, Part 1. Broth micro-dilution reference method for testing the in vitro activity of antimicrobial agents against rapidly growing aerobic bacteria involved in infectious diseases. Second edition. ISO/DIS 20776-1, Geneva, Switzerland.

Kokoska L, Kloucek P, Leuner O, Novy P. 2019. Plant-derived products as antibacterial and antifungal agents in human health care. Current Medicinal Chemistry, 26(29): 5501-5541.

Comments on the Quality of English Language

Moderate editing of English language required.

The manuscript underwent revision through Elsevier Language Editing Services. Order reference: ASLESTD1046153

Reviewer 4 Report

Comments and Suggestions for Authors

The manuscript entitled „Antibacterial, trichomonacidal, and cytotoxic activities of Pleopeltis crassinervata extracts”, submitted for evaluation to Pharmaceutics, presents the results concerning the preparation of different extracts of selected Mexican fern and evaluation of their biological effect on bacteria, parasites and kidney cell line (to evaluate its safety). Authors claim that described extracts are effective against some pathogens.

            The manuscript presents some interesting data but some comments could be useful to improve the article.

 COMMENTS TO AUTHORS

1.      I am bit confused by the lack of clear explanation of the extracts nomenclature. Names of particular extracts and related info concerning their preparation should be collected in separate table, for better clarity. According to information in the texts (262-269) and Table 2, FMeOHe and MeOH extract are related the same sample.

2.      Some abbreviations should be explained when used in the text for the first time (e.g. MZT used in line 202 but explained later.

3.      Figure 1: Subfraction Hsf2 seem to show the reverse relation between the concentration and trichomonicidal activity, in contrary to other subfractions. Could Authors comment on this observation?

4.      Line 339: Did you really mean to write “Hsf1”? Should not it be “Hsf6” instead?

Author Response

(Reviewer 4)

Response

Changes were made to the document (highlighted in yellow).

The manuscript entitled „Antibacterial, trichomonacidal, and cytotoxic activities of Pleopeltis crassinervata extracts”, submitted for evaluation to Pharmaceutics, presents the results concerning the preparation of different extracts of selected Mexican fern and evaluation of their biological effect on bacteria, parasites and kidney cell line (to evaluate its safety). Authors claim that described extracts are effective against some pathogens.

            The manuscript presents some interesting data but some comments could be useful to improve the article.

Thank you very much for the time and effort spent reviewing our manuscript. We appreciate your comments and have taken your observations into account to make the appropriate corrections to the text and improve the presentation of the document.

 COMMENTS TO AUTHORS

  1. I am bit confused by the lack of clear explanation of the extracts nomenclature. Names of particular extracts and related info concerning their preparation should be collected in separate table, for better clarity. According to information in the texts (262-269) and Table 2, FMeOHe and MeOH extract are related the same sample.

   Response:

 Due to the number of samples and their obtainment, it may cause confusion since several of the active fractions were obtained from a single extract. To improve this aspect, more information was added in the materials and methods section to clarify the nomenclature used.

  1. Some abbreviations should be explained when used in the text for the first time (e.g. MZT used in line 202 but explained later.

Response:

We apologize for not including the abbreviation previously and we thank the reviewer for pointing it. The abbreviation has been included in the M&M section (line 177).

  1. Figure 1: Subfraction Hsf2 seem to show the reverse relation between the concentration and trichomonicidal activity, in contrary to other subfractions. Could Authors comment on this observation?

Response:

The behavior of this graph could be related to the availability of the compounds in the assay, possibly due to the solubility of the compounds present or even to interactions between the phytoconstituents of the sample. By separating the mixture, the chemical environment changes, which could facilitate interactions between them, and by increasing the concentration, it is possible that the effect against the parasite is reduced, however we could not know the mechanism due to the chemical complexity of the sample.

  1. Line 339: Did you really mean to write “Hsf1”? Should not it be “Hsf6” instead?

Response:

We are wrong in that regard; what we were trying to communicate on that line was Hsf6. The corresponding correction was made.

Round 2

Reviewer 3 Report

Comments and Suggestions for Authors

Calculated and literature retention indices must be provided in the Table 4. In the methodology, GC-MS analysis should be described in better details including type of analytical column used.

Comments on the Quality of English Language

No comments.

Author Response

Thank you very much for the time and effort spent reviewing our manuscript. We appreciate your comments and have taken your observations into account to make the appropriate corrections to the text and improve the presentation of the document.

Response

Changes were made to the document (highlighted in yellow).

Comments and Suggestions for Authors:

Calculated and literature retention indices must be provided in the Table 4. In the methodology, GC-MS analysis should be described in better details including type of analytical column used.

Given that the main analytical technique is mass spectrometry, and it was also contracted as an external service, we consider that the inclusion of the retention indices calculated as reported in the literature would not provide significant additional information to our research objectives. We recognize that the exclusion of retention indices may represent a limitation in our methodological approach. However, in the literature there are numerous articles with similar methodologies, where the retention times, areas, chemical formula, and names are sufficient to report the compounds identified through databases.

Detailed information about GC-MS was added: Line 215-216